# Design and Motion Planning of a Biped Climbing Robot with Redundant Manipulator

**Qing Chang [1],\* , Xiao Luo [2], Zhixia Qiao [1] and Qian Li [1]**

[1]   School of Mechanical Engineering, Tianjin University of Commerce, No. 409 Guangrong Road,
      Beichen District, Tianjin 300134, China
[2]   School of Computer Science & Technology, Beijing Institute of Technology, No. 5 Zhongguancun South
      Street, Haidian District, Beijing 100081, China
\*   Correspondence: changq_bit@163.com; Tel.: +86-022-2667-5774

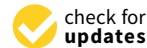

**Featured Application: A novel robot capable of performing maintenance and inspection tasks is designed to substitute humans to complete dangerous work in the railway bridges.**

**Abstract:** A novel robot capable of performing maintenance and inspection tasks for railway bridges is proposed in this paper. Termed CMBOT (climbing manipulator robot), the robot is a combination of a five-degrees-of-freedom (5-Dof) biped climbing robot with two electromagnetic feet and a redundant manipulator with 7-Dof. This capability offers important advantages for performing maintenance and inspection tasks for railway bridges. Several fundamental issues of the CMBOT, such as robotic system development and motion planning algorithms, are addressed in this paper. A series of simulations and prototype experiments were conducted to validate the proposed robotic systems and motion planning algorithm. The results of the experiments show the reliability of the robotic systems and the efficiency of the motion planning algorithm.

**Keywords:** climbing robot; redundant manipulator; robotic systems; motion planning

---

## 1. Introduction

Steel guardrails and brackets are important structures that support railway bridges and provide protection for passing trains. Continuous maintenance and inspection are required to ensure the longevity of steel structures. Hazardous conditions due to the bridge height and wind pressure from moving trains can cause difficulties in maintaining or inspecting railway bridges. To guarantee safety, workers must use heavy personal protective equipment and stop working when a train passes by. Therefore, traditional manual operation encounters many problems, such as poor security and low efficiency. Deploying special robots to replace humans can solve the problems mentioned above. The basic function of the robot is to climb bridges in a reliable and flexible way. In addition, the robot should be able to manipulate different tools to reach the surface of steel structures without collisions.

There have been several types of robots capable of climbing a wide variety of infrastructures in the past decades [1]. According to the characteristics of different infrastructures, robots have different climbing mechanisms and adhesion methods. Wheeled, tracked, legged, and combined types are the most commonly used climbing mechanisms of climbing robots [2]. Generally, robots with wheels or tracks are used on continuous surfaces with few obstacles since they have a higher speed than legged climbing robots. If the surface of an infrastructure has a complex structure or irregular obstacles, the legged climbing mechanism is more applicable than other mechanisms. Adhesion methods such as magnetic attraction [3,4], vacuum attraction [5–8], electric adhesion [9,10], gripping adhesion [11–13], and adhesive elastomer [14] have been applied in the field of climbing robots. Robots with magnetic

feet can be applied to steel structure surfaces, robots with vacuum suction can be used on smooth plane surfaces, and robots with claws or grippers are most commonly used on rough surfaces. The skid-steer mobile robot, which is a magnetic-tracked robot, can climb a vertical planar steel surface and perform welding operations [15]. Huang et al. [16] designed a tracked climbing robot with an electromagnetic array for ship inspection in shipbuilding. Equipped with probe clamping devices, the robot can perform tasks in human unfriendly environments. Inspired by rock climbers and cats, Jiang et al. [12] designed a quadruped climbing robot that can climb a rough wall surface using cross-arranged claws. Sun et al. [13] proposed a three-legged astronaut robot to assist human astronauts in space stations. Thanks to grippers with a self-locking property, the robot can move outside the space station via the special aluminum handrails freely and steady. Pagano et al. [17] designed an inchworm-inspired climbing robot with two magnetic pads for the inspection of a steel bridge and proposed a real-time motion planning method to enter the limited space inside the steel bridge. Guan et al. [18] designed a modular five-degrees-of-freedom (5-Dof) biped climbing robot with a suction module and analyzed the safety of suctions. Biped climbing robot (BiCR) was also a 5-Dof biped climbing robot with two grips. Due to the optimal collision-free grip sequence generated by a grip planning method, BiCR can climb in a complex truss environment [19].

However, the climbing robots mentioned above neglect the ability to manipulate, so their potential applications are limited to surveillance and inspection. To perform complex maintenance tasks such as rust removal and spray painting, a manipulator must be mounted on the climbing robot, which will result in problems such as body balance and motion planning. There is work that remains to add manipulating functions on the climbing robot.

Motivated by the above observation, a climbing robot with a redundant manipulator, termed as CMBOT (climbing manipulator robot), was designed to perform the inspection and maintenance tasks of railway bridges in this paper. According to the characteristics of railway bridges, the CMBOT consists of a 5-Dof biped climbing mechanism with two magnetic feet and a 7-Dof series redundant manipulator. With the redundant manipulator, the CMBOT has a superior manipulation function compared with existing climbing robots. Moreover, the CMBOT has great potential to be applied to other infrastructure by simply modifying the adhesion method and manipulation tools.

The rest of this paper is organized as follows. The robotic system of the CMBOT, including the mechanical configuration and control system, is designed in Section 2. The motion planning algorithms for the climbing mechanism and redundant manipulator are proposed and discussed in Section 3. In Section 4, the simulation and prototype experiments are conducted to verify the proposed robotic systems and related motion planning algorithms. Section 5 presents the conclusion and future work.

## 2. Design of the Robotic Systems

### 2.1. Design Indicators

From Figure 1, the guardrails and brackets of railway bridges consist of angle steel bars and round steel bars. Based on the design standard of railway bridges, most guardrails and brackets of railway bridges have the same structural parameters. However, the distance between the vertical angle steel bars may vary from 1000 mm to 1500 mm according to the construction site situation. There may also be irregular gaps between neighboring guardrails. Since every surface of the angle steel bar and round steel bar that is exposed to air should be inspected and maintained regularly, the CMBOT should have the ability to attach to the narrow surface of the steel bar, move on the steel frame with irregular gaps, and touch every surface of the steel bar. In addition, the end effector of the CMBOT should be able to manipulate maintenance tools such as a spray nozzle and rust cleaning laser.

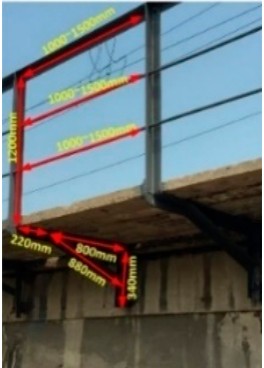 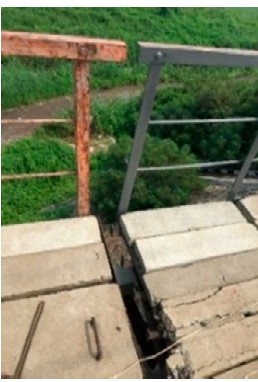

**Figure 1.** Steel guardrails and brackets in a railway bridge.

According to the abovementioned task description and analysis, the design indicators of the robot are listed as follows:

- The mass of the robot should be less than 100 kg.
- The robot can move on the guardrail and crossover a gap with a length of 10 cm.
- The robot can attach to the guardrail when it performs maintenance tasks.
- The workspace of the robot can cover a cuboid with the dimensions of $1540 \times 1100 \times 1500$ mm, and the robot is able to avoid obstacles as it performs tasks.
- The load at the end of the robot is greater than 5 kg.
- The climbing speed of the robot is not less than 0.1 m/s, the speed of the end effector is not less than 0.3 m/s.

### 2.2. Mechanical Configuration

The CMBOT consists of a 5-Dof biped climbing mechanism with two magnetic feet and a 7-Dof series redundant manipulator, as shown in Figure 2. Since the surfaces of the steel bars are too narrow to use the wheeled or tracked climbing mechanisms, the CMBOT uses the legged type of climbing mechanism. Among the existing legged climbing mechanisms that can cross irregular gaps, as shown in Figure 1, the 5-Dof biped climbing mechanism is most suitable since it has the fewest degrees of freedom and can improve the stability of motion. Five rotational joints, which are labeled J1 to J5, form the 5-Dof biped climbing mechanism. The rotational axes of the rotational joints are marked with purple arrows in Figure 2.

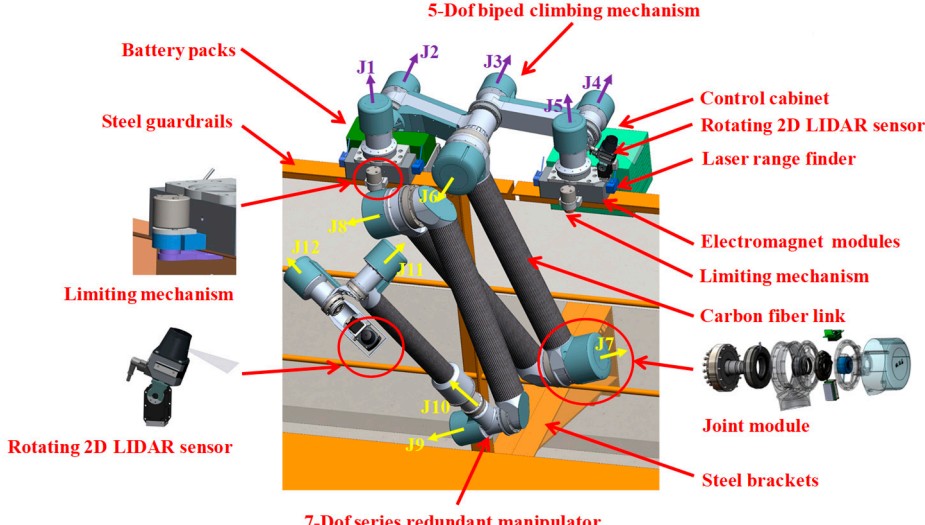

**Figure 2.** Overall structure of the climbing manipulator robot (CMBOT).

To achieve stable and accurate walking, the magnetic feet of the CMBOT consist of electromagnet modules, limiting mechanisms, and laser range finders, as shown in Figure 3. Two electromagnet modules are mounted on both ends of the climbing mechanism to ensure the CMBOT climbs on the highest horizontal angle steel bar of the guardrail. There are two laser range finders on the front and back of the electromagnet module to measure the distance between the bottom surfaces of electromagnet modules and the top surfaces of the steel guardrails. According to the distance information, J2 and J4 can adjust their angles to ensure a parallel relationship between the contact surfaces of the electromagnet modules and steel guardrails, and these actions can avoid uneven contact between the contact surfaces. In addition, limiting mechanisms are mounted on the side of both electromagnet modules. Consisting of a driven motor and limiting plate, the limiting mechanism can seize the guardrail when the magnetic feet maintain contact with the upper surface of the guardrail and release the guardrail before the electromagnet modules lift. The main objective of the limiting plate is to reduce the surface clearance between the electromagnet modules and guardrail since the surface clearance has a significant influence on the magnetic force.

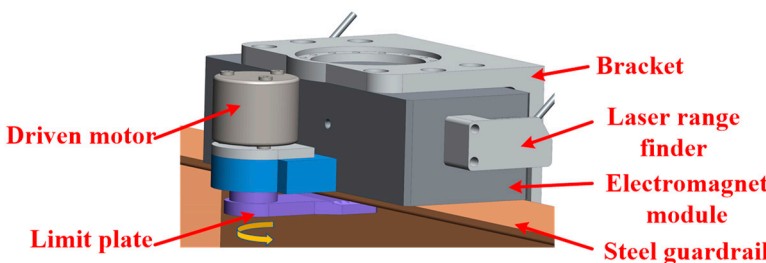

**Figure 3.** Schematic diagram of the magnetic feet.

The control cabinet and battery packs of the CMBOT are fixed on the other side of the electromagnet and can balance the weight of the manipulators. A rotating two-dimensional (2D) Light detection and ranging (LIDAR) sensor is mounted on the front of the climbing mechanism to acquire the three-dimensional (3D) characteristics used to plan the motion of the climbing mechanism. The rotating 2D LIDAR sensor, which consists of a servo motor and a 2D LIDAR, can produce a 3D point cloud of the work scene by rotating the 2D LIDAR and fusing the angle information of the servo and 2D point information of the 2D LIDAR [20]. An example of a 3D point cloud acquired by the rotating 2D LIDAR sensor is shown in Figure 4.

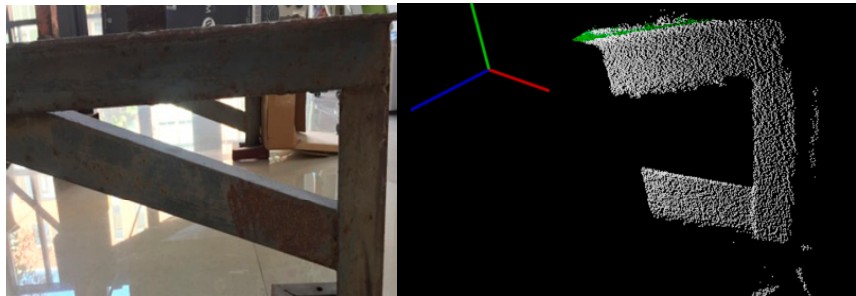

**Figure 4.** Three-dimensional (3D) point cloud of a part of the bracket acquired by the rotating two-dimensional (2D) Light detection and ranging (LIDAR) sensor.

Although the CMBOT can move on the guardrail flexibly with the climbing mechanism, the operating mechanism is still needed since the workplace is far beyond the reachable space of the climbing mechanism. Considering the characteristics of steel guardrails and brackets, a 7-Dof series redundant manipulator is used as the operating mechanism of the CMBOT. The redundant degrees of freedom can help the manipulator avoid collision with the guardrails and brackets while operating. Seven rotational joints, numbered from J6 to J12, are used for the manipulator and carbon

fiber tubes are used as the manipulator links to reduce weight. The rotational axes of the manipulator joints are marked with yellow arrows in Figure 2. The manipulator is connected to the climbing mechanism through J6, whose rotational axis is collinear with the rotational axis of J3. To ensure the torque of rotational joints is large enough for a 5-kg load, a coreless motor and harmonic reducer are used to form the joint module, as shown in Figure 5. This type of joint structure is so compact and powerful that it can produce an output torque of up to 150 Nm. There is also a rotating 2D LIDAR sensor mounted on the end of the manipulator that is used to construct the 3D scene of operating objects and obstacles before the manipulator begins maintenance tasks.

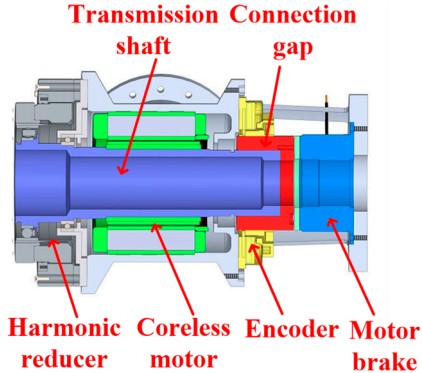

**Figure 5.** Schematic diagram of the joint module.

## 2.3. Control System

At this stage, an operator is still needed to monitor the working state of the CMBOT from the host computer in case of emergency. For a robot consisting of multiple modules such as the CMBOT, the controller of each module must be connected properly. Based on the EtherCAT BUS and CAN BUS, the distributed control architecture has been built for the CMBOT, as shown in Figure 6. A host computer is used as the top controller for human–machine interface and task management. By using the host computer, the operator can send orders such as start-stop operation or types of task to the industrial personal computer (IPC) through wireless Ethernet. Inside the control cabinet, the IPC is the core of the control system. The sensing information processing, motion planning, and actuator control are all implemented by the IPC. The IPC communicates with all 12 joint modules through the EtherCAT BUS, since it is suitable for the real-time control of the DC motor. Two microcontroller units (MCU), MCU1 and MCU2, are used to control the servo motor for the two rotating 2D LIDAR sensors and acquire the 3D point cloud of the work scene by fusing the angle information of the servo and the 2D point information from the 2D LIDAR [21]. The 3D point cloud information of the work scene is sent to the IPC for further processing through the CAN BUS. Using the rotational projection statistics (RoPS) algorithm [22], the IPC can extract the feature information of the steel guardrails and brackets that is used to plan the motion of the CMBOT. MCU3 is mainly used to control the magnetic feet of the CMBOT by acquiring the distance information of the laser range finders through `transistor-transistor logic` (TTL) and controlling the motors of the limiting mechanism and electromagnet modules through pulse width modulation (PWM). Similar to MCU1 and MCU2, MCU3 communicates with ICP through the CAN BUS.

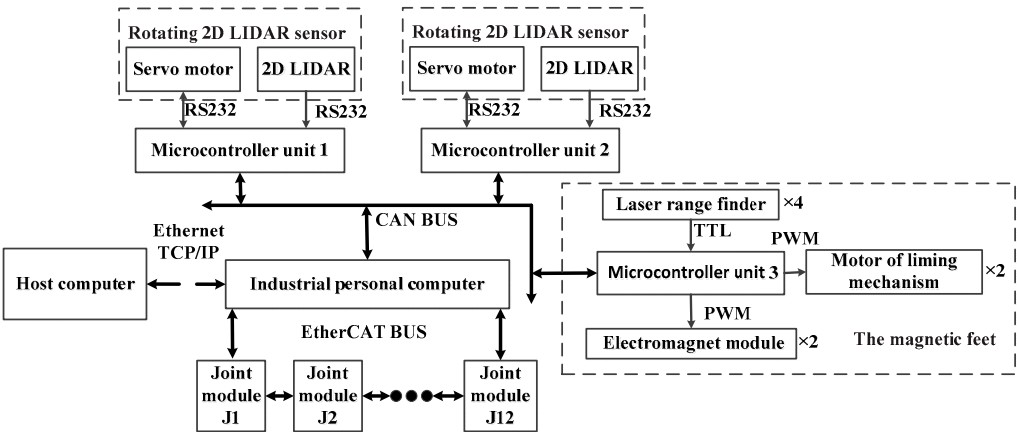

**Figure 6.** Schematic diagram of the control system.

## 2.4. Work Process

The main work procedures of the CMBOT are shown in Figure 7. Figure 8 is the work process flowchart of the CMBOT. First, the operator determines tasks and sends them to the IPC of the CMBOT through a host computer. Then, the CMBOT performs the remaining steps by itself until all the tasks are complete. The CMBOT should locate its current position suing the 3D cloud point of the guardrails acquired by the rotating 2D LIDAR sensor in the climbing mechanism. According to the relative position between the CMBOT and the guardrail, the IPC determines the working locations where the manipulator can easily perform the tasks. Then, the CMBOT moves to the nearest working location by the climbing mechanism. The end of the manipulator moves to the designated position after the CMBOT reaches the working location. The rotating 2D LIDAR sensor on the manipulator acquires the 3D point of the working surface to guide the manipulator to perform the tasks with proper tools. If the manipulator has completed the tasks at the current working location, the CMBOT will move to the next working location and the manipulator will repeat the operations mentioned above until all the tasks are complete.

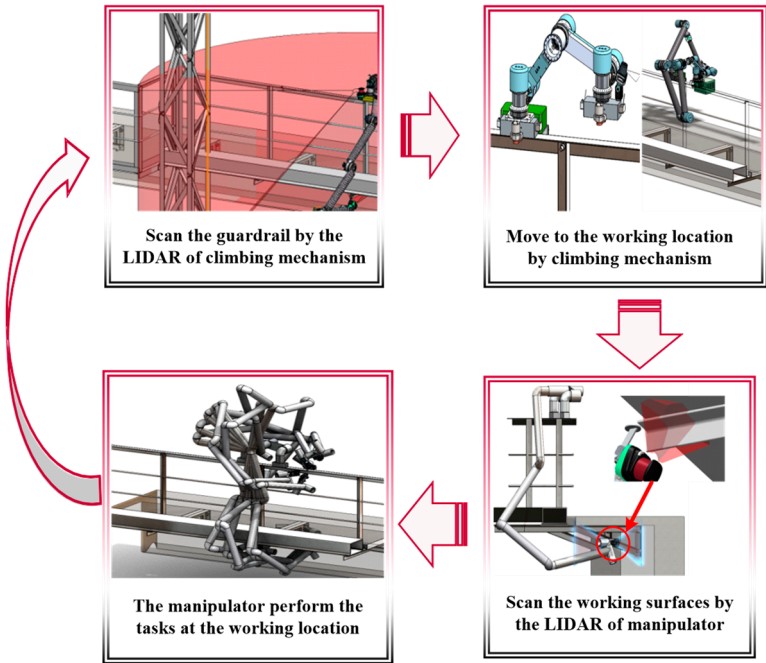

**Figure 7.** Schematic diagram of the main work procedures of the CMBOT.

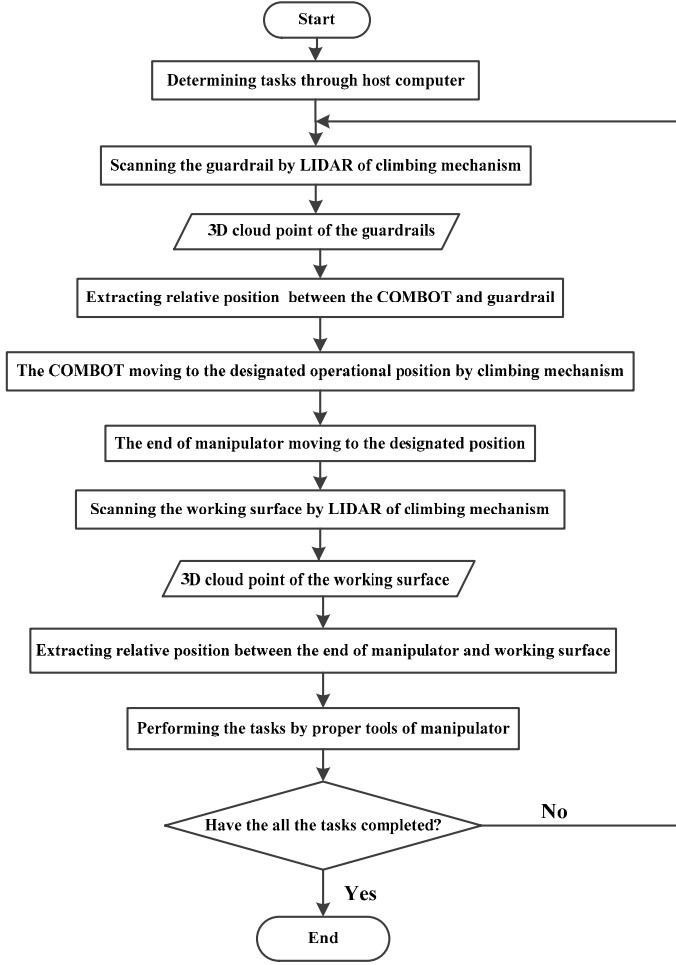

**Figure 8.** Flowchart of the work process of the CMBOT.

## 3. Motion Planning

According to the work process mentioned above, the climbing mechanism and manipulator should move in sequence, so the motion planning algorithms of the climbing mechanism and manipulator are relatively independent. The aim of climbing locomotion is to move the CMBOT to a proper working location steadily along the guardrail. Therefore, the key of the motion planning algorithm for the climbing mechanism is to ensure the balance and compliance of the motion. Since the workspace of the manipulator is so complex, collision avoidance is more of a concern for the motion planning algorithm of the manipulator.

### 3.1. Motion Planning of the Climbing Mechanism

From Figure 9, the motion of the climbing mechanism is inspired by an inchworm. A single step consists of two phases, the contraction phase and the extension phase. During the extension phase, the rear foot attaches to the surface of the handrail and the front foot extends forward along the designed trajectory until it contacts the surface. At the same time, the manipulator folds up and swings in the opposite direction of the front foot to reduce the flip torque of the rear foot. At the end of the extension phase, both the rear and front feet attach to the handrail, and the manipulator swings back to the vertical position to prepare for the contraction phase. During the contraction phase, the front foot remains in contact with the handrail and the rear foot contracts along a designed trajectory. The manipulator will also act as a counterweight to balance the CMBOT during the contraction phase.

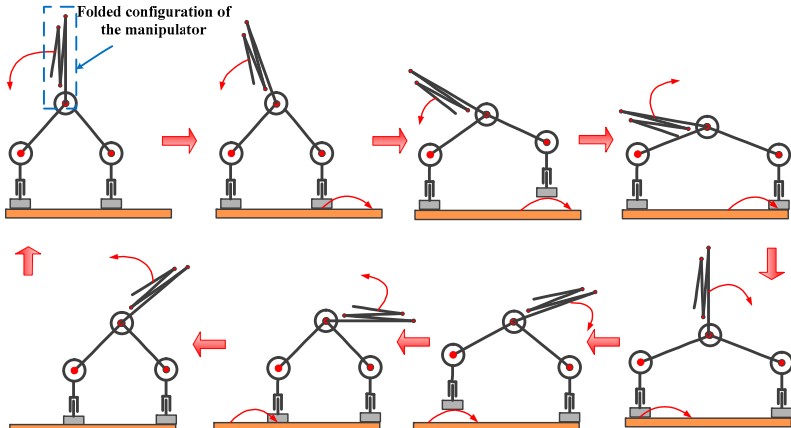

**Figure 9.** Schematic diagram of the motion of the climbing mechanism.

The climbing mechanism can be regarded as a 5-Dof manipulator when one of the feet is in contact with the handrail while the other foot is not in contact. Without loss of generality, the extension phase is analyzed in the following. Figure 10 shows the kinematic model of the climbing mechanism which is built using the Denavit–Hartenberg (D–H) convention method [23]. Coordinate (B) is the base coordinate of the climbing mechanism, which is also the workspace coordinate of the climbing mechanism. Coordinate (*i*) (*i* = 1, 2, 3, 4, 5, 6) represents the coordinate of J*i*. The Z-axis in Figure 10 is defined as the rotation axis of the joints (the Z-axis of J2, J3, J4 are vertical to paper and the positive direction faces outward). The Z-axis of J6, which connects the climbing mechanism, is opposite to the Z-axis with J3. Defined by the D–H method, $\theta_i$ is the rotation angle of J*i*. Point P is located at the center of the lower surface of the front foot and can be regarded as the end of the 5-Dof manipulator. The blue trajectory in Figure 10 is the trajectory of point P, and $p = (p_x, p_y, p_z)^T$ is the displacement vector of P that is represented in coordinate (B). In Figure 10, $H$ is the height of the trajectory, $S$ is the displacement of the trajectory along the X-axis, and $\varphi_y$ and $\varphi_z$ represent the angle the front foot rotates about the Y-axis and Z-axis of coordinate (B).

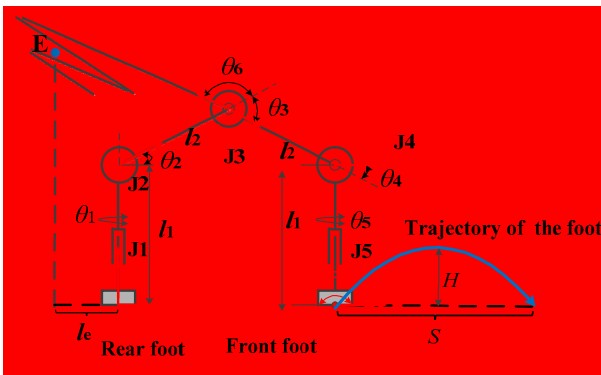

**Figure 10.** Kinematic model of the climbing mechanism.

The D–H parameters are shown in Table 1. According to the D–H method, vector $p$ that expresses the position of P relative to coordinate (B) can be represented by the following equation:

$$p = T_1^B T_2^1 T_3^2 T_4^3 T_5^4 e_P, \tag{1}$$

where, $\mathbf{e}_p = (0\ 0\ 0\ 1)^T$ is the position of point P relative to coordinate (5), and $T_j^i$ represents the homogenous transformation matrix from $\{i\}$ to $\{j\}$ and can be expressed by the following equation:

$$T_j^i = \begin{bmatrix} \cos(\theta_i) & -\sin(\theta_i) & 0 & a_i \\ \sin(\theta_i)\cos(\alpha_i) & \cos(\theta_i)\cos(\alpha_i) & -\sin(a_i) & -\sin(a_i)d_i \\ \sin(\theta_i)\sin(\alpha_i) & \cos(\theta_i)\sin(\alpha_i) & \cos(\alpha_i) & \cos(\alpha_i)d_i \\ 0 & 0 & 0 & 1 \end{bmatrix}. \tag{2}$$

**Table 1.** Denavit–Hartenberg (D–H) parameters of the climbing mechanism.

| Joint | Link Torsional Angle $\alpha_i$ (°) | Link Length $a_i$ (mm) | Link Offset $d_i$ (mm) | Joint Angle $\theta_i$ (°) |
|---|---|---|---|---|
| 1 | 0 | 0 | 0 | $\theta_1$ |
| 2 | 90 | 0 | $l_1$ | $\theta_2$ |
| 3 | 0 | $l_2$ | 0 | $\theta_3$ |
| 4 | 0 | $l_2$ | 0 | $\theta_4$ |
| 5 | 90 | 0 | $l_1$ | $\theta_5$ |

Passing the parameters of Table 1 to Equation (1), $p$ can be expressed explicitly by the following equation:

$$p_x = \cos(\theta_1)(l_2\cos(\theta_2) + l_2\cos(\theta_2 + \theta_3) + l_1\sin(\theta_2 + \theta_3 + \theta_4)); \tag{3}$$

$$p_y = \sin(\theta_1)(l_2\cos(\theta_2) + l_2\cos(\theta_2 + \theta_3) + l_1\sin(\theta_2 + \theta_3 + \theta_4)); \tag{4}$$

$$p_z = l_1 + l_2\sin(\theta_2) + l_2\sin(\theta_2 + \theta_3) - \cos(\theta_2 + \theta_3 + \theta_4). \tag{5}$$

$\varphi_y$ and $\varphi_z$ can be expressed by the following equation:

$$\varphi_y = -\theta_2 + \theta_3 + \theta_4; \tag{6}$$

$$\varphi_z = \theta_1 + \theta_5. \tag{7}$$

Inverse kinematics is used to calculate the joints angles when the designed trajectory is given. To prevent collisions between the magnetic modules and the handrail, $\varphi_y$ should remain at 0° during the locomotion. The sum of $\theta_2$, $\theta_3$, and $\theta_4$ should be 0 according to Equation (6). Passing this constraint to Equations (3)–(5), the values of $\theta_1$, $\theta_3$, $\theta_2$, $\theta_4$, and $\theta_5$ can be calculated successively as:

$$\begin{cases} \theta_1 = \arctan\left(\frac{p_y}{p_x}\right) \\ \theta_3 = \arccos\left(\frac{(p_y)^2 + (p_z\cos(\theta_1))^2 - 2(l_2\cos(\theta_1))^2}{2(l_2\cos(\theta_1))^2}\right) \\ \theta_2 = \frac{\theta_3}{2} - \arcsin\left(\frac{p_z}{\sqrt{(p_x)^2 + (p_y)^2}}\right) \\ \theta_4 = \theta_2 - \theta_3 \\ \theta_5 = \varphi_y \end{cases} \tag{8}$$

To prevent the CMBOT from falling, the manipulator rotates along the Z-axis of J6 and acts as a counterweight to balance the CMBOT. From Figure 10, the folded configuration of the manipulator rotates about J6 to reduce the torque about the Y-axis of (B). Point B is the origin of (B), the magnitude of the torque about B due to the gravity of climbing mechanism is given by:

$$M = \sum_{i=1}^{5} \vec{BL_i} \times m_i g + \sum_{i=1}^{5} \vec{BL_i} \times m_i a_{L_i}, \tag{9}$$

where, M is the torque about *B*, $L_i$ is the center of mass of link *i*, $m_i$ is the mass of link *i*, $a_{L_i}$ is the acceleration of point $L_i$, and g is the magnitude of gravity acceleration. Then, the angle of Joint 6 can be calculated as:

$$\theta_6 = M_y / l_e, \tag{10}$$

where, $M_y$ represents the torque about the *Y*-axis of (B), and $l_e$ represents the distance between the equivalent center of mass of manipulator E and *Y*-axis of (B).

The trajectories of the magnetic feet also play an important role in motion stability. A low-contact impact trajectory is proposed to reduce the influence of impingement against the handrail of the climbing configuration. Based on the compound cycloid, the proposed trajectory is smooth, so there is not a sudden change in the velocity and acceleration of the trajectory, and both the velocity and acceleration will be equal to 0 when the magnetic feet contact the handrail. The equations of the trajectory are given by:

$$\begin{cases} \Delta p_x = S\left(\frac{t}{T_b} - \frac{1}{2\pi}\sin\left(2\pi\frac{t}{T_b}\right)\right) 0 \le t < T_b \\ \Delta p_z = \begin{cases} 2H\left(\frac{t}{T_b} - \frac{1}{4\pi}\sin\left(4\pi\frac{t}{T_b}\right)\right) 0 \le t < \frac{T_b}{2} \\ 2H\left(1 - \frac{t}{T_b} + \frac{1}{4\pi}\sin\left(4\pi\frac{t}{T_b}\right)\right) 0 \le t < \frac{T_b}{2} \end{cases} \\ \Delta p_y = D\left(\frac{t}{T_b} - \frac{1}{2\pi}\sin\left(2\pi\frac{t}{T_b}\right)\right) 0 \le t < T_b \end{cases}, \tag{11}$$

where, $\Delta p = (\Delta p_x, \Delta p_y, \Delta p_z)^T$ is the relative displacement vector of the trajectory which is represented in (B). In Figure 10, *H* is the height of the trajectory and *S* is the displacement of the trajectory along the *X*-axis. *D* is the displacement of the trajectory along the *Y*-axis, which is not shown in Figure 10. *t* is the time of the motion and $T_b$ is the period of the contraction phase and the extension phase.

### 3.2. Motion Planning of the Redundant Manipulator

A 7-Dof redundant manipulator was selected so the redundant degree of freedom can play an important role in avoiding collisions between the manipulator and obstacles. A planning algorithm based on the gradient projection method is proposed in this section. The kinematic model of the manipulator is built using the D–H method, as shown in Figure 11. Coordinate (K) is the base coordinate of the manipulator and coordinate (*i*) (*i* = 7, 8, 9, 10, 11, 12) represents the coordinate of J*i*. The Z-axis in Figure 11 is defined as a rotation axis of joints. Point Q is located on the end of the manipulator and its trajectories are planned according to the tasks. The D–H parameters are shown in Table 2. According to the D–H method, vector $q = (q_x \ q_y \ q_z)^T$, which expresses the position of Q relative to coordinate (K), can be represented by the following equation:

$$q = T_6^K T_7^6 T_8^7 T_9^8 T_{10}^9 T_{11}^{10} T_{12}^{11} e_q, \tag{12}$$

where $e_q = (0 \ 0 \ -l_7 \ 1)^T$ is the position of point Q relative to coordinate {12}, $T_i{}^j$ represents the homogenous transformation matrix from {*i*} to {*j*}.

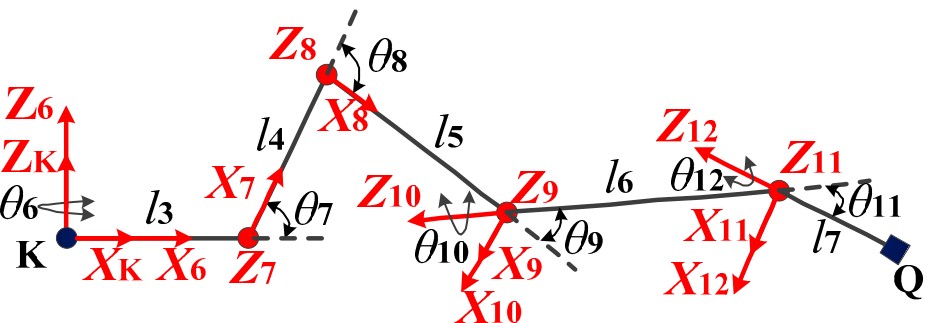

**Figure 11.** Kinematic model of the manipulator.

**Table 2.** D–H parameters of the manipulator.

| Joint | Link Torsional Angle $\alpha_i$ (°) | Link Length $a_i$ (mm) | Link Offset $d_i$ (mm) | Joint Angle $\theta_i$ (°) |
|---|---|---|---|---|
| 6 | 0 | 0 | 0 | $\theta_6$ |
| 7 | −90 | $l_3$ | 0 | $\theta_7$ |
| 8 | 0 | $l_4$ | 0 | $\theta_8$ |
| 9 | 0 | $l_5$ | 0 | $\theta_9$ |
| 10 | 90 | 0 | $l_6$ | $\theta_{10}$ |
| 11 | −90 | 0 | 0 | $\theta_{11}$ |
| 12 | 90 | 0 | 0 | $\theta_{12}$ |

According to the mathematical introduction of robotic manipulation [24], the generalized velocity of the end of the manipulator and the angular velocity of the joints are connected as:

$$v = J\beta, \tag{13}$$

where, $v$ denotes the generalized velocity of the end of the manipulator in (K), $\beta = \left[\dot{\theta}_6\ \dot{\theta}_7\ \dot{\theta}_8\ \dot{\theta}_9\ \dot{\theta}_{10}\ \dot{\theta}_{11}\ \dot{\theta}_{12}\right]^{\mathrm{T}}$ denotes the angular velocity of the manipulator in its joint space, $J \in \mathrm{R}^{6\times7}$ denotes the Jacobian matrix of the manipulator. Generalized velocity $v$ includes the linear velocities and angular velocities along the axes of (K) and can be represented as $v = \left(\dot{q}_x\ \dot{q}_y\ \dot{q}_z\ \omega_x\ \omega_y\ \omega_z\right)^{\mathrm{T}}$. The Jacobian matrix $J$ can be calculated according to Equation (12), and the general solution to Equation (13) is given by the following equation [25]:

$$\beta = J^+ v + \left(I - J^+ J\right)Z, \tag{14}$$

where $J \in \mathrm{R}^{7\times6}$ is the pseudoinverse matrix of $J$ and can be denoted as $J^+ = J^{\mathrm{T}}\left(J\ J^{\mathrm{T}}\right)^{-1}$. $J^+ v$ is the minimum norm solution of $\beta$, which can guarantee that the manipulator tracks the planned trajectory with a minimum sum of squares of the angular velocity of the joints. $I \in \mathrm{R}^{7\times7}$ is the identity matrix. Furthermore, $(I - J^+ J)Z$ is a homogeneous solution of $J^+ v$ and can adjust the configuration of the manipulator to meet special demands under the premise of unchanged trajectory. $Z$ is the optimizing index that contains the variable of $\beta$. The resultant joint angular velocity can be regarded as a combination of the least solution of the minimum norm and a homogeneous solution created by the action of a projection operator $(I - J^+ J)$, which describes the redundancy of the system, mapping an arbitrary $\beta$ into the null space of the transformation. By applying various functions of $\beta$ to compute vector $Z$, the manipulator can be reconfigured to achieve a desired secondary criterion under the constraint of the specified end-effector velocity.

In general, if the minimum distance between the manipulator and an obstacle is less than the safety threshold, the danger of collision will increase significantly. To reduce the possibility of collision, the manipulator should move in the opposite direction of obstacles and increase the minimum distance until it exceeds the safety threshold. It is necessary to determine the minimum distance and direction vector between the manipulator and obstacles. However, calculating the minimum distance between the manipulator and obstacles is difficult, since the shapes of the manipulator and obstacles are irregular. To solve this problem, the manipulator and obstacles are simplified as a combination of regular shapes based on their structural characteristics, as shown in Figure 12. The manipulator is simplified as a set of spheres that can cover all surfaces of itself. The climbing mechanism and railway bridge are simplified as a set of cuboids and act as the obstacles to the manipulator.

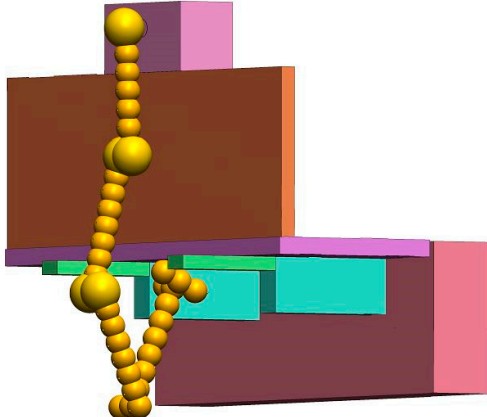

**Figure 12.** Manipulator and railway bridges simplified as spheres and cuboids.

Since spheres and cuboids are typical convex shapes, the minimum distance between them can be calculated by the Gilbert–Johnson–Keerthi (GJK) algorithm [26]. Unlike many other distance algorithms, the geometry data does not have to be stored in a specific format. Let $C_i$ be labels for $i$th virtual spheres and $D_j$ be the labels for $j$th virtual cuboids. The minimum distance between $C_i$ and $D_j$ and the closest points in $C_i$ and $D_j$ can be calculated efficiently based on the GJK algorithm. The distance between $C_i$ and $D_j$ is denoted by $d_{ij}$, and the unit direction vector that points from the closest point in $D_j$ to the closet point in $C_i$ is denoted by $u_{ij}$ (see Figure 13).

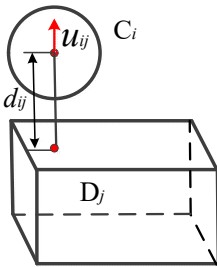

**Figure 13.** Schematic diagram of minimum distance between virtual spheres and obstacles.

Let $d_s$ be the safety threshold during locomotion. If $d_{ij}$ is large enough to avoid the danger of a collision, the manipulator will not execute the obstacle avoidance algorithm. In contrast, if $d_{ij}$ is smaller than $d_s$, the manipulator should change configuration to avoid obstacle $D_j$. In the meantime, the most effective way to avoid the obstacle in this situation is to move $C_i$ away from $D_j$ in the direction of $u_{ij}$. Then, the related velocity can be calculated by:

$$s_{ij} = \begin{cases} \mathbf{0} \ d_{ij} > d_s \\ \frac{v_{\max}}{2}\left(\cos\left(\pi \frac{d_{ij}}{d_s}\right) + 1\right)u_{ij} \ d_{ij} \le d_s \end{cases}. \tag{15}$$

In Equation (15), $s_{ij}$ denotes the escape velocity of sphere $C_i$ in the direction of $u_{ij}$ and $v_{\max}$ denotes the maximum escape velocity. Additionally, the escape velocity will be generated when $d_{ij} \le d_s$ and increase to the maximum with the decrease of $d_{ij}$.

If $s_{ij} \ne 0$, sphere $C_i$ should escape $D_i$ at a speed of $s_{ij}$ until $s_{ij}$ becomes 0. The escape motion should have the following equation:

$$s_{ij} = J_{C_i} \beta, \tag{16}$$

where $J_{C_i}$ is the Jacobian of the center of sphere $C_i$. If there are $k(k > 1)$ pairs of spheres and cuboids whose minimum distances are less than $d_s$, the related escape velocity and related Jacobian can be renamed $s_k$ and $J_k$. Then, we can obtain the following matrix equation according to Equation (16):

$$\begin{bmatrix} s_1 \\ s_2 \\ \vdots \\ s_k \end{bmatrix} = \begin{bmatrix} J_1 \\ J_2 \\ \vdots \\ J_k \end{bmatrix} \beta. \tag{17}$$

Let $v^*$ represent $[s_1 \ s_2 \cdots s_k]^{\mathrm{T}}$ and $J^*$ represent $[J_1 \ J_2 \cdots J_k]^{\mathrm{T}}$. Substituting Equation (14) into Equation (17) yields:

$$v^* = J^* J^+ v + J^* \left( \mathbf{I} - J^+ J \right) Z. \tag{18}$$

The solution that increases the minimum obstacle distance is provided by the pseudoinverse, given by:

$$Z = \left[ J^* \left( \mathbf{I} - J^+ J \right) \right]^+ \left( v^* - J^* J^+ v \right). \tag{19}$$

Substituting Equation (16) back into Equation (13) to determine the solution to avoid obstacles and track the end effector at the same time, the following equation is obtained:

$$\beta = J^+ v + \left( \mathbf{I} - J^+ J \right) \left[ J^* \left( \mathbf{I} - J^+ J \right) \right]^+ \left( v^* - J^* J^+ v \right). \tag{20}$$

Each term in Equation (17) has an explicit physical interpretation. $J^+ v$ can guarantee the end of the manipulator tracks the desired trajectory with the minimum joint velocity norm. $(\mathbf{I} - J^+ J)[J^*(\mathbf{I} - J^+ J)]^+$ is used to transform the desired motion of spheres from Cartesian space to the joint space using the pseudoinverse. $v^* - J^* J^+ v$ describes the desired velocity of spheres, and $v^*$ is the escape velocity, which is calculated based on the minimum distance. $J^* J^+ v$ is the velocity as a result of satisfying the end effector velocity constraint.

## 4. Experiments

To verify the effectiveness of the proposed robotic system and related motion planning algorithms, a series of simulations and experiments with the CMBOT were conducted. As shown in Figure 14, a robot motion simulation system for the CMBOT was developed to examine the robot trajectories for obvious problems before the trajectories are transferred to the actual robot system, thus ensuring the safety of the robot motion. The motion planning algorithms of the climbing mechanism and manipulator were simulated using the proposed simulation software.

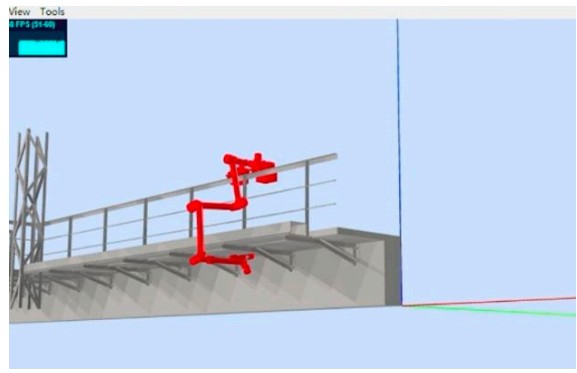

**Figure 14.** Snapshot of the motion simulation system for the CMBOT.

When planning the motion of the climbing mechanism, the following foot trajectory parameters were selected: $S = 240$ mm, $H = 50$ mm, $D = 0$, $T_b = 10$ s, $\varphi_y = 0$, $\varphi_z = 0$. According to Equation (8),

the curves of displacement, velocity, and acceleration of the low-contact impact trajectory during the extension phase are shown in Figure 15. As shown in Figure 15, the proposed trajectory is smooth so there is not a sudden change in the velocity or acceleration of the trajectory, and both the velocity and acceleration will be equal to 0 when the magnetic feet contact the handrail. This is very important for reducing the impact when the magnetic feet contact the handrail. The joint angles of the climbing mechanism during the motion are shown in Figure 16.

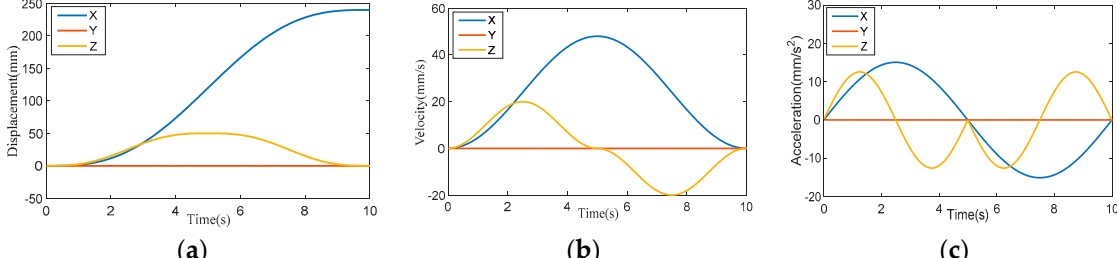

**Figure 15.** Curves of the low-contact impact trajectory: (**a**) Displacement, (**b**) Velocity, and (**c**) Acceleration.

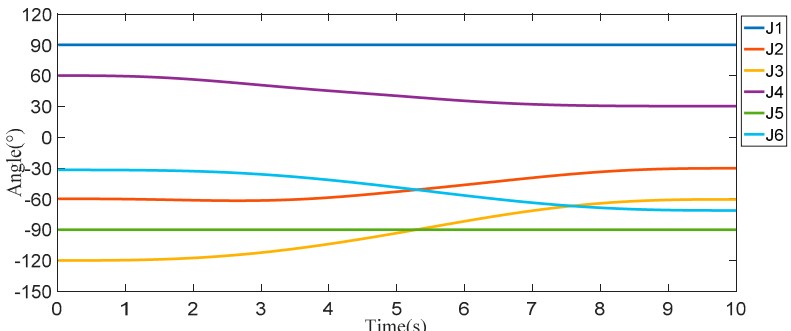

**Figure 16.** The joint angles of the climbing mechanism.

The proposed motion planning algorithm of the redundant manipulator was also verified in the simulation. One of the most commonly used trajectories will be shown in the simulation. The end of the manipulator will track this selected trajectory from the start point to the side of the bracket. The position and pose angle of the end of the manipulator at the start point is (0 mm, 820 mm, $-2790$ mm, $0°$, $0°$, $-90°$)$^\text{T}$ in coordinate (6). The simulation will last 20 s, and equations of the desired trajectory can be calculated by:

$$\dot{q}_\text{x} = \begin{cases} 80t - 450, & 0 \le t < 5 \\ 0, & 5 \le t < 20 \end{cases} \tag{21}$$

$$\dot{q}_\text{y} = \begin{cases} -160, & 0 \le t < 5 \\ -78, & 5 \le t < 20 \end{cases} \tag{22}$$

$$\dot{q}_\text{z} = \begin{cases} 98t, & 0 \le t < 5 \\ 0, & 5 \le t < 20 \end{cases} \tag{23}$$

$$\omega_\text{x} = \begin{cases} 7.34t - 18.35, & 0 \le t < 5 \\ 0, & 5 \le t < 20 \end{cases} \tag{24}$$

$$\omega_\text{y} = \begin{cases} 28.8t, & 0 \le t < 5 \\ 0, & 5 \le t < 20 \end{cases} \tag{25}$$

$$\omega_\text{z} = \begin{cases} 28.8t, & 0 \le t < 5 \\ 0, & 5 \le t < 20 \end{cases} \tag{26}$$

According to Equations (21)–(26), the curves of position and pose angle of the end of the manipulator in the simulation are shown in Figures 17 and 18. In the first 5 s of the simulation, the end of the manipulator move forms the start point to working position along the spatial curve, and then move along a straight line at the root of the bracket.

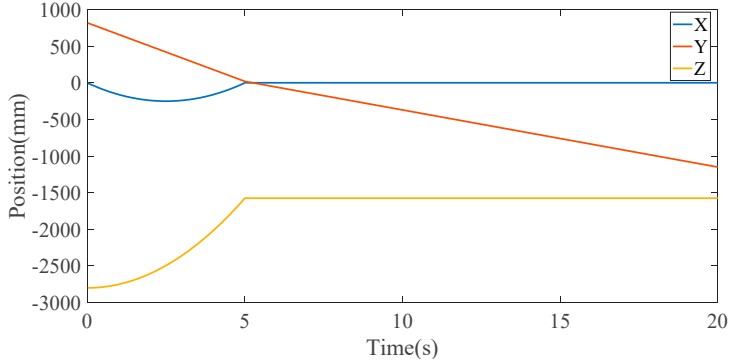

**Figure 17.** The position of the end of the manipulator in the simulation.

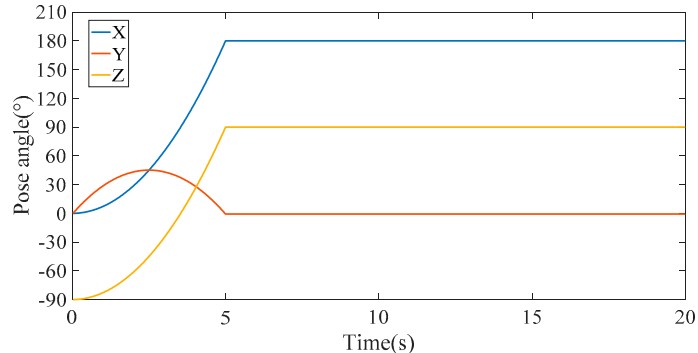

**Figure 18.** The pose angle of the end of the manipulator in the simulation.

The information of the environment should be acquired before we calculate the joint angles of the redundant by the proposed motion planning algorithm. Firstly, with the help of the long manipulator, the LIDAR sensor on the manipulator can scan the working surfaces from some certain positions where it is far enough from the obstacles of the railway bridge. The 3D point data will be transformed into coordinate (6) (i.e., the base of the manipulator) according to the manipulator configuration. Then 3D point cloud from different visual angles will be combined to obtain more complete information about the bridge. The combined 3D point data needs to be filtered to remove clutter and make the data be distributed evenly before further processing. Figure 19 shows the processed 3D point cloud of the railway bridge. Though we have acquired the standard 3D model of bridge construction drawings, it is very time-consuming and inefficient to match the standard 3D model of bridge with the obtained 3D point cloud. Five sub models were extracted from the standard 3D model of bridge, and the coordinate origins of the models were denoted as feature points (FP), as shown in Figure 20. The distribution of the 3D point cloud in the sub model is unique and can be recognized easily. Then, we recognized the sub models in the obtained 3D point cloud by using the rotational projection statistics (RoPS) algorithm [22], the results are the position vectors of the feature points in coordinate (6). The feature points construct the skeleton of the railway bridge in the coordinate system of the obtained 3D point cloud, and the boxes surfaces envelope the skeleton to form the simplified model of the railway bridge.

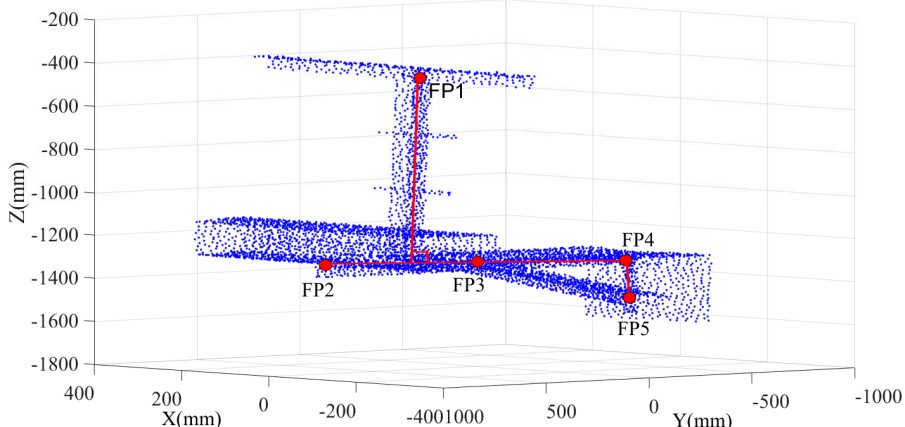

**Figure 19.** Schematic diagram of the processed 3D point cloud.

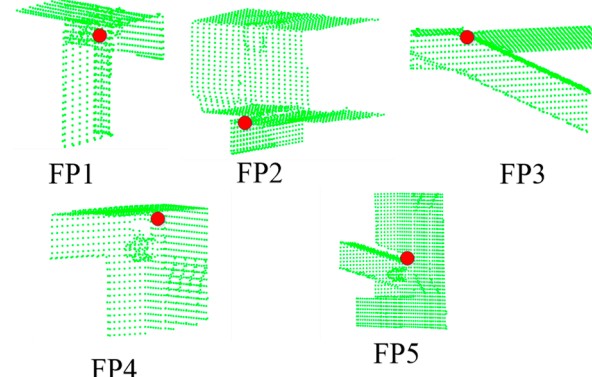

**Figure 20.** Schematic diagram of the sub models and related feature points.

Figure 21 shows the schematic diagram of the motion during the simulation. In the figure, the boxes represent obstacles such as the handrail and bracket of the railway bridge, the cluster of blue straight-line segments denote the manipulator links, the red curve with circles denotes the trajectory of the end of the manipulator. As shown in the figure, using the proposed motion planning algorithm, the end of the manipulator can move with the designed trajectory without colliding with any of the obstacles. The joint angles of the redundant manipulator during the motion are shown in Figure 22.

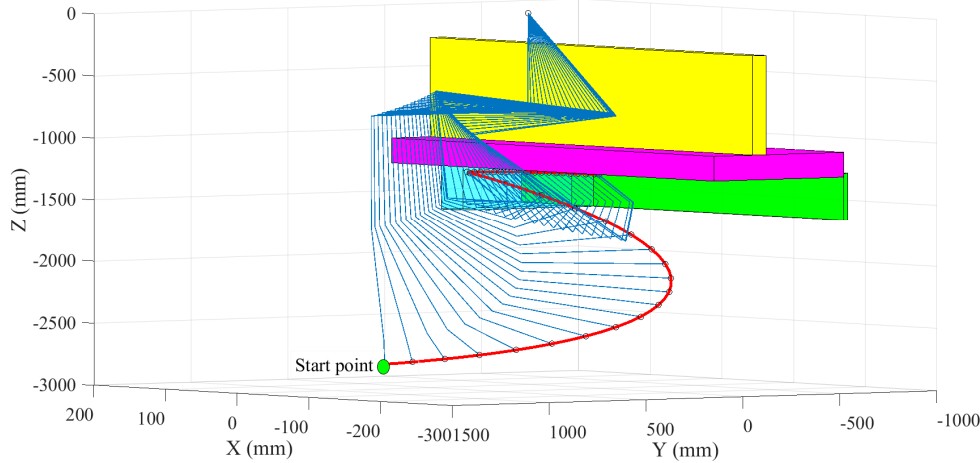

**Figure 21.** Schematic diagram of the motion of the manipulator.

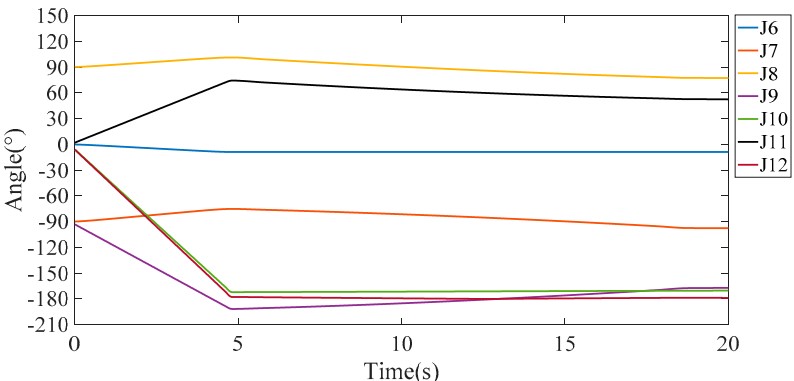

**Figure 22.** The joint angles of the redundant manipulator.

The movement must be checked by the simulation system before the robot starts to work. If some part of the manipulator is too close to an obstacle in the simulation, the operator will adjust the position the base of the manipulator by the climbing mechanism. We must ensure safety before any operation is carried out. After a series of simulations, the proposed motion planning algorithms were tested on a prototype. The prototype of the CMBOT was built based on the design analysis mentioned above, and an indoor experiment platform with the same structure as the railway bridge was constructed. The joint angles of the climbing mechanism in the experiment are shown in Figure 23. It can be seen from Figures 16 and 23 that the expected and actual angles are basically consistent. Snapshots of the extension phase of the climbing mechanism experiment are shown in Figure 24. The rear foot of the climbing mechanism attaches to the surface of the handrail and the front foot extends forward along the designed trajectory until it contacts the surface. At the same time, the manipulator folds up and swings in the opposite direction of the front foot to reduce the flip torque of the rear foot. At the end of the extension phase, both the rear and front feet are attached to the handrail, the manipulator swings back to the vertical position to prepare for the contraction phase.

Figure 25 shows joint angles of the redundant manipulator in the experiment. It can be seen from Figures 22 and 25 that the expected and actual angles of the redundant manipulator are also basically consistent. We can conclude from the results that the torque and control precision of the proposed joint modules are good enough to apply for the CMBOT. Snapshots of the redundant manipulator experiment are shown in Figure 26. The end of the redundant manipulator moved along the designed trajectory. Considering there should be a certain distance between the maintenance tools and the maintenance surfaces, the distance was set to 20 cm in the experiment. Snapshots of the maintenance experiment are shown in Figure 26.

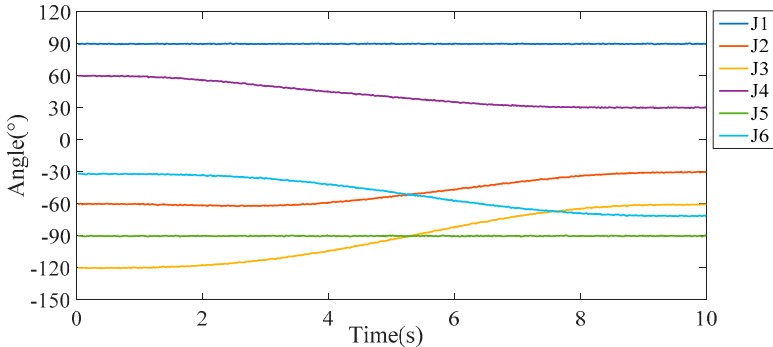

**Figure 23.** The joint angles of the climbing mechanism in the experiment.

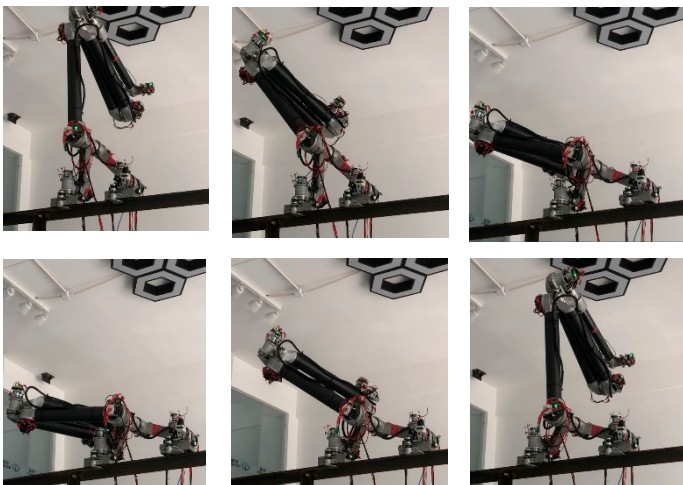

**Figure 24.** Snapshots of the extension phase of the climbing mechanism experiment.

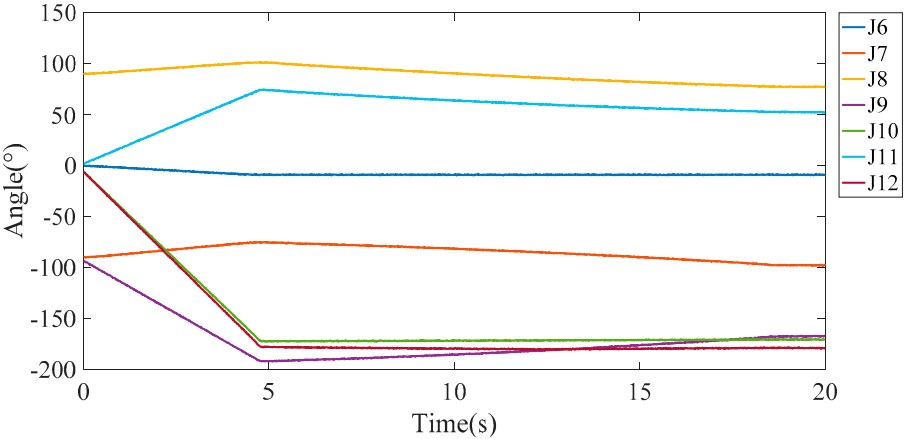

**Figure 25.** The joint angles of the redundant manipulator in the experiment.

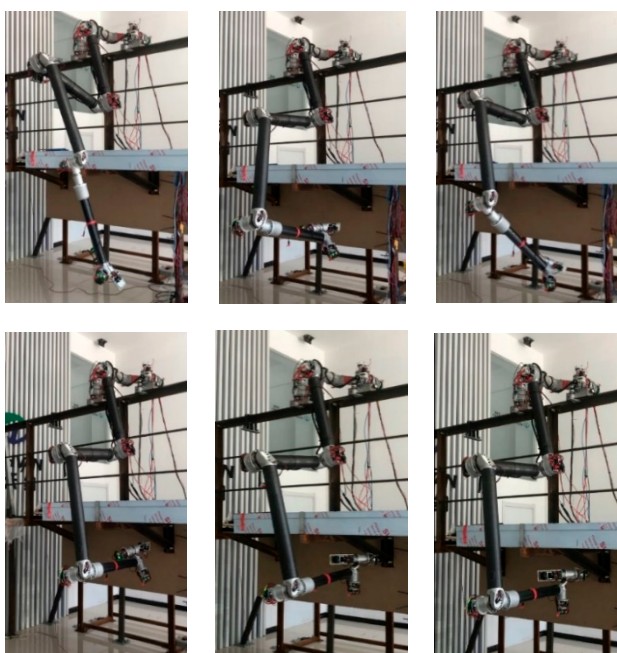

**Figure 26.** Snapshots of the redundant manipulator experiment.

## 5. Discussion and Conclusions

Based on the analysis of the inspection and maintenance tasks of railway bridges, a robot named CMBOT was proposed in this paper. The CMBOT consists of a 5-Dof biped climbing mechanism and a 7-Dof series redundant manipulator. The mechanical configuration and control system of the CMBOT were introduced in this paper. Due to the novel mechanical configuration, the CMBOT has a superior manipulation function to other existing climbing robots. Moreover, the CMBOT has great potential to be applied to other infrastructure by simply modifying the adhesion method and manipulation tools. The motion planning algorithms of the climbing mechanism and the redundant manipulator were also introduced. The aim of the motion planning algorithm of the climbing mechanism is to ensure the balance and compliance of the motion. Collision avoidance is more of a concern for the motion planning algorithm of the manipulator. The simulation and prototype experiments were used to verify the effectiveness of the proposed robotic system and related motion planning algorithms.

There is some remaining work before the CMBOT can be applied to an actual railway bridge. For example, maintenance tools such as a spray nozzle and rust cleaning laser will be mounted on the redundant manipulator and experiments and analysis of the maintenance effect of CMBOT will be performed using the experiment platform.

**Author Contributions:** Conceptualization, Q.C.; Funding acquisition, Q.C. and X.L.; Methodology, Q.C.; Project administration, X.L.; Software, X.L.; Validation, Z.Q. and Q.L.; Writing—original draft, Q.C.; Writing—review & editing, Z.Q. and Q.L.

**Funding:** This research was funded by National Key Research and Development Program of China (2016YFC0803005), National Natural Science Foundation of China (61501034) and Enterprise Technology Commissioner Foundation of Tianjin (18JCTPJC64400).

**Conflicts of Interest:** The authors declare no conflict of interest.

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
