# Peer review of "Design and Motion Planning of a Biped Climbing Robot with Redundant Manipulator"

_applsci, doi:10.3390/app9153009_

Reviewer 1 Report

Dear Authors

 Major criticism:

Please add grid lines in the graph (Figure 17. Schematic diagram of the motion of the manipulator)

Experimental research results can be added (The joint angles of the redundant manipulator)

Author Response

Point 1: Please add grid lines in the graph (Figure 17. Schematic diagram of the motion of the manipulator).

Response 1: Thanks for the reviewer’s comment. We have added grid lines in Figure 17 (figure number now change to 21 in this revised manuscript) and this change were highlighted in the revised manuscript.

Point 2: Experimental research results can be added (The joint angles of the redundant manipulator)

Response 2: Thanks for the reviewer’s suggestion. We have added the joint angles of climbing mechanism (Figure 23) and redundant manipulator (Figure 25) in the experiment in the revised manuscript and the and more information about the simulation and experiment are also supplied in the revised manuscript.

Reviewer 2 Report

The paper presents a new robotic system for maintenance of railway bridges. The robot climbs on a supporting guardrails and uses robotic arms to inspect the bridge or perform simple tasks like painting. The robot is well designed and the whole work seems to be motivated by a real applications.

The paper is written in a nice way and easily understandable; the pictures are also made well.

The motion planning depends on the 3D models of the robot (which is available as a set of geometric primitives (spheres)) and

and the environment. The paper however does not describe in detail how is the map of the environment (i.e., the map of bridge and its parts) obtained.

This relates to other questions, that are not described sufficiently:

How is the 3D point cloud processed? How are new 3D point data integrated/matched/used to update an already existing 3D model of the environment?

How are the geometric primitives (boxes) of the 3D model obtained from the point cloud?

Regarding motion planning of the manipulator, the pose of the manipulator is computed using inverse kinematics, but it is not clear, how the desired trajectory (i.e., position of the end-effector) is calculated?

Also, authors propose to calculate minimal distance between the manipulator and the obstacles and move the manipulator away from the obstacles, if some part of the manipulator is too close to an obstacle. What happens, if this movement is not possible without significantly reconfiguration of the robot (i.e., when the 'escaping' trajectory needs to pass a singularity)?

I would like to hear why authors do not employ some general planning methods, e.g. sampling-based planners like RRT, which are often used for manipulators and which could avoid the situation when the singularity need to be passed.

Author Response

Point 1: The motion planning depends on the 3D models of the robot (which is available as a set of geometric primitives (spheres)) and the environment. The paper however does not describe in detail how is the map of the environment (i.e., the map of bridge and its parts) obtained.

Response 1: Thanks for the reviewer’s comment. The map of the environment is acquired by the 3D point data. Firstly, we find some feature points through 3D point cloud processing. Then, the feature points construct the skeleton of the railway bridge. And finally, the boxes surfaces envelope the skeleton to form the simplified model of the railway bridge. 

The related introduction and schematic diagrams have been added in the revised manuscript.

Point 2: How is the 3D point cloud processed? How are new 3D point data integrated/matched/used to update an already existing 3D model of the environment?

How are the geometric primitives (boxes) of the 3D model obtained from the point cloud?

Response 2: Thanks for the reviewer’s comment. Firstly, with the help of the long manipulator, the LIDAR sensor on the manipulator can scan the working surfaces from some certain positions where is far enough from the obstacles of the railway bridge. The 3D point data will be transformed into coordinate {6} (i.e., the base of the manipulator) according to the manipulator configuration. Then 3D point cloud from different visual angles will be combined to obtain more complete information about the bridge. The combined 3D point data needs to be filtered to remove clutter and make the data distributed evenly before further processing. Though we have acquired the standard 3D model of bridge from the construction drawings, it is very time-consuming and inefficient to match the standard 3D model of bridge with the obtained 3D point cloud. Five sub models are extracted from the standard 3D model of bridge, and the coordinate origins of the sub models are denoted as feature points. The distribution of 3D point cloud in the sub model is unique and can be recognized easily. Then we recognized the sub models in obtained 3D point cloud by using rotational projection statistics (RoPS) algorithm (reference 22), the results are the position vectors of the feature points in coordinate {6}. The feature points construct skeleton of the railway bridge in the coordinate system of obtained 3D point cloud, and the boxes surfaces envelope the skeleton to form the simplified model of the railway bridge.

The related introduction and schematic diagrams have been added in the latest revised manuscript.

Point 3: Regarding motion planning of the manipulator, the pose of the manipulator is computed using inverse kinematics, but it is not clear, how the desired trajectory (i.e., position of the end-effector) is calculated?

Response 3: Thanks for the reviewer’s comment. We have added the equations (Equations 21-26) and figures (Figure 17 and Figure 18) of the desired trajectory in the revised manuscript.

Point 4: Also, authors propose to calculate minimal distance between the manipulator and the obstacles and move the manipulator away from the obstacles, if some part of the manipulator is too close to an obstacle. What happens, if this movement is not possible without significantly reconfiguration of the robot (i.e., when the 'escaping' trajectory needs to pass a singularity)?

Response 4: Thanks for the reviewer’s comment. Fist, the CMBOT is specially designed for the maintenance and inspection tasks for railway bridges, the structural parameters of manipulator have been optimized to reduce the probability of singularity near the working trajectories and keep the proper distance from the known obstacles which can be obtained from the construction drawings of railway bridges. Second, the movement must be checked by the simulation system before the robot start to work. If some part of the manipulator is too close to an obstacle in the simulation, the operator will adjust the position the base of manipulator by the climbing mechanism. We must ensure safety before any operation is carried out.

Point 5: I would like to hear why authors do not employ some general planning methods, e.g. sampling-based planners like RRT, which are often used for manipulators and which could avoid the situation when the singularity need to be passed.

Response 5: Thanks for the reviewer’s suggestion. Sample-based planners such as RRT and PRM are widely used to plan collision-free paths for high-Dof robots, they are often fast and can avoid the singularity. But standard sampled-based planners compute piecewise linear paths that can be executed precisely by stopping the robot at every vertex along the paths, the jerky and unnatural paths will reduce efficiency of the robot and cause vibration at the end of manipulator. The links of the manipulator are so long that the vibration have an important influence in the trajectory precision. In future, we will use some improved RRT algorithm in the proposed robotic system by smoothing the jerky paths.

Round  2

Reviewer 2 Report

Authors improved the paper according the previous suggestions; they explained (and wrote in the paper) details about data processing, 3D object matching and motion planning, they updated images.